

# From standard wind measurements to spectral characterization: turbulence length scale and distribution

Mark Kelly[1]

[1]Wind Energy Department, Risø Lab./Campus, Danish Technical University; Roskilde 4000, Denmark.

*Correspondence to:* Mark Kelly (`MKEL@DTU.DK`)

**Abstract.**

In wind energy, the the effect of turbulence upon turbines is typically simulated using wind 'input' time-series based on turbulence spectra. The velocity components' spectra are characterized by the amplitude of turbulent fluctuations, as well as the length scale corresponding to the dominant eddies. Following the IEC standard, turbine loads calculations commonly

involve use of the Mann spectral-tensor model to generate timeseries of the turbulent three-dimensional velocity field. In practice, this spectral-tensor model is employed by adjusting its three parameters: the dominant turbulence length scale $L_{\mathrm{MM}}$ (peak length scale of an undistorted isotropic velocity spectrum), the rate of dissipation of turbulent kinetic energy $\varepsilon$, and the turbulent eddy-lifetime (anisotropy) parameter $\Gamma$. Deviation from 'ideal' neutral sheared turbulence—i.e. for non-zero heat flux and/or heights above the surface layer—is, in effect, captured by setting these parameters according to observations.

Previously, site-specific $\{L_{\mathrm{MM}}, \varepsilon, \Gamma\}$ were obtainable through fits to measured three-dimensional velocity component spectra recorded with sample rates resolving the inertial range of turbulence ($\gtrsim 1\,\mathrm{Hz}$); however, this is not feasible in most industrial wind energy projects, which lack multi-dimensional sonic anemometers and employ loggers that record measurements averaged over intervals of minutes. Here a form is derived for the shear dependence implied by the eddy-lifetime prescription within the Mann spectral-tensor model, which leads to derivation of useful forms of the turbulence length scale. Subsequently

it is shown how $L_{\mathrm{MM}}$ can be calculated from commonly-measured site-specific atmospheric parameters, namely mean wind shear ($dU/dz$) and standard deviation of streamwise fluctuations ($\sigma_u$). The derived $L_{\mathrm{MM}}$ can be obtained from standard (10-minute average) cup anemometer measurements, in contrast with an earlier form based on friction velocity.

The new form is tested across several different conditions and sites, and is found to be more robust and accurate than estimates relying on friction velocity observations. Assumptions behind the derivations are also tested, giving new insight

into rapid-distortion theory and eddy-lifetime modelling—and application—within the atmospheric boundary layer. The work herein further shows that distributions of turbulence length scale, obtained using the new form with typical measurements, compare well with distributions $P(L_{\mathrm{MM}})$ obtained by fitting to spectra from research-grade sonic anemometer measurements for the various flow regimes and sites analyzed. The new form is thus motivated by and amenable to site-specific probabilistic loads characterization.



# 1 Introduction

Of the atmospheric parameters which are generally input into (or required by) wind turbine loads calculation codes, several stand out, due to their prominence in load contributions. These are the 'mean' wind speed $U$, the standard deviation of streamwise turbulent velocity $\sigma_u$, the shear exponent $\alpha$ (calculated from wind speeds at multiple heights, e.g. Kelly et al., 2014a), and

the characteristic turbulence length scale corresponding to the most energetic turbulent motions (e.g. Wyngaard, 2010).

Within the context of obtaining site-dependent statistics of the most relevant load-driving parameters from typical or standard wind measurements, we focus upon the relevant parameter which is most difficult to mesaure: the 'Mann-model length scale' $L_{\mathrm{MM}}$ (Mann, 1994); i.e., the turbulence length scale, which corresponds to the 'energy-containing sub-range' of turbulence which contributes most to turbulent velocity fluctuations (and also turbine loads).

Independent derivation of some eddy-lifetime relations has also been done concurrently by de Mare and Mann (2016), towards creation of a model for time-varying eddy lifetime.

Dimitrov et al. (2017) found that both fatigue and extreme turbine loads can be sensitive to $L_{\mathrm{MM}}$ (as well as $\Gamma$), in addition to the dominant influences of mean wind speed $U$ and streamwise turbulence 'strength' $\sigma_u$. This is also consistent with the earlier finding of Sathe et al. (2013), whom found that stability could affect fatigue loads

# 2 Theory

## 2.1 Eddy lifetime

A number of forms exist to estimate eddy lifetime $\tau_e$, though these can be generally expressed as the ratio of a length scale (taken as the reciprocal of wavenumber, $k^{-1}$) to a velocity scale which follows from some integrated form of the (scalar) kinetic energy spectrum $E(k)$:

$$\tau_e \sim k^{-p-1}\left[\int_k^\infty \kappa^{-2p} E(\kappa)d\kappa\right]^{-1/2},\tag{1}$$

where the characteristic velocity scale can be generically described by

$$k^p\left[\int_k^\infty \kappa^{-2p} E(\kappa)d\kappa\right]^{1/2}.$$

Comparing to the 'coherence-destroying diffusion time' of Comte-Bellot and Corrsin (1971) and to the reciprocal of eddy-damping rates from Lesieur (1990), for use with rapid-distortion theory Mann (1994) chose an eddy lifetime that depends on eddy size (wavenumber) according to

$$\tau_M \propto k^{-1}\left[\int_k^\infty E(\kappa)d\kappa\right]^{-1/2};\tag{2}$$




i.e., equivalent to $p = 0$ in terms of (1). The choice (2) for eddy lifetime was found to behave more reasonably than both the Comte-Bellot and Corrsin (1971) 'diffusion time' (where $p = 1$)[1], as well as the timescale $[k^3 E(k)]^{-1/2}$ (which in the inertial range is equivalent to $p = -1$)[2] implicit in eddy-damped quasi-normal Markovian [EDQNM] models (Andre and Lesieur, 1977; Lesieur, 1990); both of the latter lifetime models do not (reliably) integrate to give finite $\sigma_u^2$.

Mann (1994) re-writes $\tau_M$ as

$$\tau_M(k) = \frac{\Gamma}{dU/dz} \frac{(kL_{\mathrm{MM}})^{-2/3}}{\sqrt{{}_2F_1\left(\frac{1}{3}, \frac{17}{6}; \frac{4}{3}; \frac{-1}{(kL_{\mathrm{MM}})^2}\right)}}, \tag{3}$$

where ${}_2F_1$ is Gauss' hypergeometric function (Abramowitz and Stegun, 1972)[3]. The eddy lifetime definition (3) is used in practical implementation of the spectral tensor model (e.g. Mann, 2000), and it notably defines a parameter of this model: the eddy lifetime scaling parameter $\Gamma$, also known as the anisotropy factor. The Mann (1994) spectral-tensor model employs rapid

distortion theory ('RDT') with an initial turbulent kinetic energy spectrum of the isotropic von Kármán form

$$E_{\mathrm{vK}}(k) = \alpha \varepsilon^{2/3} L_{\mathrm{MM}}^{5/3} \frac{(kL_{\mathrm{MM}})^4}{[1 + (kL_{\mathrm{MM}})^2]^{17/6}}, \tag{4}$$

with $\alpha = 1.7$ (von Kármán, 1948). Using (4) in the proportionality expression (2) produces

$$\tau_{M|E \to E_{\mathrm{vK}}} = \frac{c_\tau k^{-2/3}}{\sqrt{\frac{3}{2}\alpha\varepsilon^{2/3}\,{}_2F_1\left(\frac{1}{3}, \frac{17}{6}; \frac{4}{3}; \frac{-1}{(kL_{\mathrm{MM}})^2}\right)}}, \tag{5}$$

where we have introduced the coefficient $c_\tau$ to equate (2) with (3). Then we have an expression relating the Mann-model

parameters to the shear $dU/dz$:

$$\Gamma = \frac{c_\tau}{\sqrt{3\alpha/2}} \frac{dU}{dz} L_{\mathrm{MM}}^{2/3} \varepsilon^{-1/3}. \tag{6}$$

Now $\tau_M$ can be seen to depend upon $k$, $L_{\mathrm{MM}}$, and $\varepsilon$. The eddy-lifetime can be reduced and clarified via ${}_2F_1\left\{\frac{1}{3}, \frac{17}{6}; \frac{4}{3}, (-kL_{\mathrm{MM}})^{-2}\right\} \simeq [1 + 3.07(kL_{\mathrm{MM}})^{-2}]^{-1/3}$ to give the more understandable von Kármán-like form[4]

$$\tau_M(k; L_{\mathrm{MM}}, \varepsilon) \simeq \frac{0.82 c_\tau}{\sqrt{\alpha\varepsilon^{2/3}}} k^{-2/3} \left[1 + \frac{3.07}{(kL_{\mathrm{MM}})^2}\right]^{1/6}. \tag{7}$$

The expression (6) can be made yet more useful to relate the turbulent length scale to measureable parameters, as shown in section 2.2.

---

[1] The Mann (1994) expression is also equivalent (or at least proportional) to the 'convection time' of Comte-Bellot and Corrsin (1971).

[2] The reciprocal of eddy-damping rate, $[k^3 E(k)]^{-1/2}$, is equal in the inertial range to (1) with $p = -1$ since $E(\kappa) \to \kappa^{-5/3}$ there. This expression is also similar to the 'rotation time' or 'strain time' given by Comte-Bellot and Corrsin (1971), but it should be noted that such expressions integrate from 0 to $k$, i.e. over eddies larger than $1/k$.

[3] The hypergeometric function ${}_2F_1\left[\frac{1}{3}, \frac{17}{6}; \frac{4}{3}; -(kL_{\mathrm{MM}})^{-2}\right]$ approaches 1 for $kL_{\mathrm{MM}} \gg 1$ (the inertial range), and simplifies to $a_{HG}(kL_{\mathrm{MM}})^{2/3}$ for $kL_{\mathrm{MM}} \ll 1$, where $a_{HG} \equiv (3\sqrt{\pi}/4) f_\Gamma(4/3)/f_\Gamma(17/6) \simeq 0.69$ and $f_\Gamma(x)$ is the Euler-gamma function.

[4] Note $\sqrt{2/3} \simeq 0.82$, and $3.07 = a_{HG}^{-3}$; c.f. footnote 3. In (7), $\alpha\varepsilon^{2/3}$ is kept together for comparison with (4), and because $\alpha\varepsilon^{2/3}$ is commonly used as an input to the spectral-tensor model instead of $\varepsilon$ (e.g. Mann et al., 2002; IEC 61400–1, Edition 3, 2005).



### 2.1.1 Eddy lifetime and equilibrium

The parameters $\{\varepsilon, \Gamma, L_{\mathrm{MM}}\}$ are site-dependent, and in practice have been obtained from measurements through fits of the model output to observed spectra (Mann, 2000), relying on (at least three of) $F_{11}$, $F_{13}$, $F_{33}$, and $F_{22}$ (e.g. Sathe et al., 2013; Dimitrov et al., 2017). The model starts with an (undistorted) isotropic incompressible turbulence spectral tensor

$$\Phi_{ij}(\mathbf{k}) = \frac{\delta_{ij}k^2 - k_i k_j}{4\pi k^4} E(k) \tag{8}$$

where $E(k)$ is taken to be $E_{\mathrm{vK}}(k)$ following (4), then the $\Phi_{ij}$ are distorted—i.e. the rapid-distortion equations are solved—per (three-dimensional) wavenumber over a time $\tau_M(k)$ via rapid-distortion theory ['RDT'].

The rapid-distortion equations discussed here do not explicitly solve for production of normal stresses (which sum to twice the turbulent kinetic energy) nor shear stress, though they do include (Fourier-transformed) terms for perturbing isotropic stresses[5] to account for the anisotropic effect of a constant shear $dU/dz$; however, RDT does *not* include dissipation (e.g. Pope, 2000). Instead, in the spectral-tensor model the dissipation rate $\varepsilon$ is a parameter giving the amplitude of the undistorted (initial) spectrum via (4); $\varepsilon$ can also be obtained via fits of precalculated Mann-model output to measured (distorted) spectra, so $\varepsilon$ effectively gives the inertial-range amplitudes of the distorted velocity component spectra, which have been distorted for a time $\tau_M(k)$. The parameter $\Gamma$ serves as a factor that determines the amount of distortion and associated anisotropy— and thus separation between the peaks of the different component spectra (the $ww$-spectrum peak is at higher wavenumbers than $vv$-peak, which is at higher $k_1$ than the $uu$ spectral peak); increasing $\Gamma$ (Mann, 1994) corresponds to more anisotropy. Thus a stationary equilibrium result is achieved via the eddy-lifetime prescription together with rapid-distortion of an isotropic spectrum—with inertial-range amplitudes and also $\tau_M$ both depending on $\varepsilon$ (via eqns. 3 and 6)—whereby shear-production of TKE is in effect balanced by dissipation. The resultant shear stress $\langle uw \rangle$ (expressible now in terms of $\varepsilon$) can be multiplied by $2\partial U/\partial z$ to give the implied production rate of $\langle uu \rangle$, which with $vv$ and $ww$ (through $\Gamma$) gives the implied TKE production rate, amounting to $P = \varepsilon$; such an equilibrium, enforced by $\tau_M$, can also be inferred from de Mare and Mann (2016).

## 2.2 Characteristic length scale

Noting that the spectrum of a variable integrates to the variance of said variable, then invoking (8) with the isotropic von Kármán form (4) for $E(k)$ and exploiting $F_{11}(k_1) = \iint \Phi_{11} dk_2 dk_3$, one obtains the isotropic streamwise turbulence variance

$$\sigma_{\mathrm{iso}}^2 = 2\int_0^\infty \frac{9}{55}\alpha\varepsilon^{2/3}L_{\mathrm{MM}}^{5/3}[1+(k_1 L_{\mathrm{MM}})^2]^{-5/6}dk_1$$
$$= 0.69\alpha\varepsilon^{2/3}L_{\mathrm{MM}}^{2/3} \tag{9}$$

which is the undistorted streamwise variance. The factor 0.69 is the numerical value of $\frac{9}{55}\sqrt{\pi}f_\Gamma(\frac{1}{3})/f_\Gamma(\frac{5}{6})$ and $f_\Gamma(x)$ is the Euler gamma function (Abramowitz and Stegun, 1972, see also footnote 3 above). Then using (9) in (6) we get a relation for

---

[5]Assuming a constant mean shear $dU/dz$, the spectral-tensor model solves (Fourier-transformed versions of) rapid-distortion equations for streamwise normal stress $\langle u_1 u_1 \rangle$ and shear stress $\langle u_1 u_3 \rangle$; multiplying these by $dU/dz$ one obtains the corresponding production rates: $P_{11} = -2\langle u_1 u_3 \rangle dU/dz$ and $P_{13} = -\langle u_3 u_3 \rangle dU/dz$ (Pope, 2000, Ch.11).





the isotropic (undistorted) turbulence length scale implied by the lifetime-model (3),

$$L_{\mathrm{MM}} \simeq \left( \frac{1.5\Gamma}{c_\tau} \right) \frac{\sigma_{\mathrm{iso}}}{dU/dz},\tag{10}$$

where the leading term in parenthesis is expected to be of order 1.

### 2.2.1 Relation to observations

Peña Diaz et al. (2010) suggested that the Mann-model length scale is proportional to the classic mixing length $\ell_* \equiv u_*/(dU/dz)$ multiplied by an empirical constant, i.e.

$$L_{\mathrm{MM}} = c_m \ell_* = \frac{c_m u_{*,\mathrm{obs}}}{dU/dz}\tag{11}$$

where they assign $c_m = 1.7$. However, we find from observations that on average $c_m \approx 2.3$ over flat land, i.e. $\langle L_{\mathrm{MM}}/\ell_* \rangle = 2.3$ (see next section). Combining (10)–(11) one sees that $c_\tau$ decreases with the relative magnitude of measured shear stress
(as $\sigma_{\mathrm{iso}}/u_{*,\mathrm{obs}}$); this is also expressed usefully through the measured ratio of streamwise fluctuation amplitude to friction velocity:

$$c_\tau \simeq \frac{1.5\Gamma}{c_m} \frac{\sigma_{\mathrm{iso}}}{u_{*,\mathrm{obs}}} = \frac{1.5\Gamma}{\sigma_{\mathrm{u,obs}}/\sigma_{\mathrm{iso}}} \left( \frac{\sigma_{\mathrm{u,obs}}/u_{*,\mathrm{obs}}}{c_m} \right).\tag{12}$$

From the above and (10) one subsequently then finds

$$L_{\mathrm{MM}} \simeq \frac{\sigma_{u,\mathrm{obs}}}{dU/dz} \left( \frac{c_m}{\sigma_{\mathrm{u,obs}}/u_{*,\mathrm{obs}}} \right).\tag{13}$$

For constant $(\sigma_{\mathrm{u,obs}}/u_{*,\mathrm{obs}})$, (13) implies that the turbulence scale $L_{\mathrm{MM}}$ can be expressed *independently* of $\Gamma$, given $\sigma_{u,\mathrm{obs}}$ and $dU/dz$.

Caughey et al. (1979) reported the mean profile of $\sigma_u^2(z)$ from the seminal 'Kansas experiment', showing that $(\sigma_u/u_*)_0^2 \approx 5$–6 in the homogeneous atmospheric surface layer (their Fig. 5). The corresponding value of $(\sigma_u/u_*)_0$ is approximately 2.3; thus, if $c_m \approx 2.3$ as well, then (12) reduces to

$$c_\tau \approx \frac{1.5\Gamma}{\sigma_{u,\mathrm{obs}}/\sigma_{\mathrm{iso}}}.\tag{14}$$

Given the definition of $c_\tau$ through (6), $c_\tau$ is a constant; since (9) shows $\sigma_{\mathrm{iso}}$ is independent of $\Gamma$, then $\sigma_{u,\mathrm{obs}} \propto \Gamma$. Consistent with this argument, (13) reduces to

$$L_{\mathrm{MM}} \approx \frac{\sigma_{u,\mathrm{obs}}}{dU/dz}.\tag{15}$$

Using (15), $L_{\mathrm{MM}}$ can simply be diagnosed from typical measurements, e.g. 10-minute average cup-anemometer output, at two
(or more) heights. The length $L_{\mathrm{MM}}$ can also be cast in terms of variables commonly used in wind engineering, notably the turbulence intensity $I_u$ and shear exponent $\alpha$. Invoking $dU/dz = \alpha U/z$ (Kelly et al., 2014a) and defining $I_{\mathrm{obs}} \equiv \sigma_{u,\mathrm{obs}}/U$, then (15) becomes $L_{\mathrm{MM}} \approx z I_{\mathrm{obs}}/\alpha$.





### 2.2.2 Modelled spectra: covariances, anisotropy and Γ

The spectral Mann-model ('MM') distorts the isotropic von Kármán spectral tensor ($\Phi_{ij}(\mathbf{k})$, eq. 4), per wavenumber via rapid-distortion theory over the wavenumber-dependent eddy-lifetime $\tau_M$, such that the component spectra become anisotropic at wavenumbers outside (lower than) the inertial range; the degree of distortion—and thus anisotropy—are consequently rep-resented by the eddy-lifetime parameter Γ. Above we showed via mixing-length arguments that $L_{\mathrm{MM}}$ is independent of Γ, resulting in (15). Possible Γ-dependences can also be examined by considering the shear stress

$$\langle uw \rangle_{\mathrm{MM}} = -u_{*,\mathrm{MM}}^2 = 2 \int\limits_0^\infty F_{13}(k_1) dk_1 \tag{16}$$

obtained from the modelled spectral tensor component $F_{13}(k_1) = \iint \Phi_{13} dk_2 dk_3$, which is expected to be a function of Γ. Indeed Mann (1994, Figure 4) shows this to be the case, with modeled stress $\langle uw \rangle_{\mathrm{MM}}/\sigma_{\mathrm{iso}}^2$ varying almost linearly between 0 and $-1$ for $0 < \Gamma < 5$; then $u_{*\mathrm{MM}}^2/\sigma_{\mathrm{iso}}^2 \approx \Gamma/5$. Subsequently from (12) one has

$$c_\tau \approx \frac{1.5\sqrt{5\Gamma}}{c_m} \frac{u_{*\mathrm{MM}}}{u_{*,\mathrm{obs}}} \approx \frac{0.64\Gamma}{u_{*,\mathrm{obs}}/\sigma_{\mathrm{iso}}} \tag{17}$$

for $c_m \approx 2.3$, in analogy with (14); thus we expect $u_{*,\mathrm{obs}} \propto \Gamma$, similar to the expected behavior of $\sigma_{u,\mathrm{obs}} \propto \Gamma$ following (14).

In addition to the approximate expression (17), which is based on the simplified relation $u_{*\mathrm{MM}}^2/\sigma_{\mathrm{iso}}^2 \approx \Gamma/5$, it is possible to derive an exact relation based on the the Mann-model shear stress (16); but this is cumbersome and analytically intractable. Though de Mare and Mann (2016) derived implicit expressions toward relating $\{\Gamma, dU/dz, L_{\mathrm{MM}}\}$ to the eddy lifetime and integral of the modelled stress spectrum (16), these must be evaluated numerically or graphically. An explicit expression corresponding to $c_m^{-1} = \ell_*/L_{\mathrm{MM}}$ (like eqn. 11 here) was derived by de Mare and Mann (2016), but it depends on numerically integrating the stress spectrum.

As spectra fitted to Mann-model outputs correspond to distorted *anisotropic* turbulence, and noting the Γ-dependence of $u_{*\mathrm{MM}}$ discussed above, we expect $\sigma_{u,\mathrm{MM}}$ to also depend on Γ. From Figure 4 of Mann (1994) we find $\sigma_{u,\mathrm{MM}}^2/\sigma_{\mathrm{iso}}^2 \simeq (1 + 0.14\Gamma^2)$, which for $\Gamma \gtrsim 2$, the range corresponding to ABL observations (e.g. Sathe et al., 2013), becomes roughly $\sigma_{u,\mathrm{MM}} \approx \sigma_{\mathrm{iso}}(0.61 + 0.3\Gamma)$.

### 2.3 Ideal, neutral surface-layer implications

Within the atmospheric surface-layer ('ASL'), in the homogeneous stationary (ideal) limit under neutral conditions, $dU/dz \to u_*/(\kappa z)$ so that (11) reduces to $L_{\mathrm{MM}} \to c_m \kappa z \approx 0.92z$. Similarly, in this 'log-law regime' $\varepsilon_{\mathrm{ASL,N}} = u_*^3/(\kappa z)$, so that (6) becomes $\Gamma_{\mathrm{ASL,N}} = c_\tau(3\alpha/2)^{-1/2}(L_{\mathrm{MM}}/\kappa z)^{2/3}$ or equivalently $L_{\mathrm{MM}}|_{\mathrm{ASL,N}} = (3\alpha/2)^{3/4}\kappa z[\Gamma/c_\tau]^{3/2}$ which via (12) can be written

$$\begin{aligned} L_{\mathrm{MM}}|_{\mathrm{ASL,N}} &= \left(\frac{3\alpha}{2}\right)^{3/4} \kappa z \left[\frac{c_m u_{*,\mathrm{obs}}}{1.5\sigma_{\mathrm{iso}}}\right]^{3/2} \\ &\simeq 1.1\kappa z \left[\frac{c_m}{\sigma_{u,\mathrm{obs}}/u_{*,\mathrm{obs}}} \frac{\sigma_{u,\mathrm{obs}}}{\sigma_{\mathrm{iso}}}\right]^{3/2} \end{aligned} \tag{18}$$





Thus for $c_m = \sigma_{u,\mathrm{obs}}/u_{*,\mathrm{obs}}$, we see that the Mann (1994) eddy-lifetime formulation (3) implies $L_{\mathrm{MM}} \to 1.1\kappa z(\sigma_{u,\mathrm{obs}}/\sigma_{\mathrm{iso}})^{3/2}$ in the neutral ASL. Meanwhile, as noted just above, the mixing-length form (11) implies $L_{\mathrm{MM}} \to c_m\kappa z$; this is consistent with (18) under the condition that $(\sigma_{u,\mathrm{obs}}/\sigma_{\mathrm{iso}}) \simeq (c_m/1.1)^{2/3}$ or roughly $\sigma_{u,\mathrm{obs}} \approx 1.6\sigma_{\mathrm{iso}}$ for $c_m \simeq 2.3$.

## 3  Observations and results

Since the choice of eddy lifetime form (3) leads to a shear-dependent relation (6) between the spectral-tensor model parameters, one obtains (10) for the undistorted (isotropic) length scale, with $L_{\mathrm{MM}} \propto \sigma_{\mathrm{iso}}(dU/dz)^{-1}$; further invoking a mixing-length argument then leads to a relation (15) for $L_{\mathrm{MM}}$ in terms of quantities that are directly measureable via standard wind-industry (one-dimensional cup) anemometers. Here we test (15) as well the assumptions leading to it, through measured wind speed, shear, and turbulent velocity component spectra. We also find a form for the distribution of $L_{\mathrm{MM}}$ over all conditions—as would
be needed in practice to represent the turbulence length scales of flows experienced by wind turbines at a given site.

The spectra are measured via three-dimensional sonic anemometers on the primary meteorological mast located at the Danish National Test Site for Large Wind Turbines (Høvsøre), 1.75 km from the western coast of Denmark (Mann et al., 2005; Peña Diaz et al., 2016). The anemometers give 20 Hz samples of all three velocity components and temperature[6] at heights of 10, 20, 40, 60, 80, and 100 m. This allows calculation of mean speeds, directions, and vertical shear of mean speed over individual 10-minute records; in particular we focus on heights of $z = 20$ m and $z = 80$ m, as we are able to calculate shear at
(across) these heights using the measurements at 10, 40, 60, and 100 m, while also using the measured wind speed components and susbequent spectra at $z = \{20, 80\}$ m. The parameters $\{L_{\mathrm{MM}}, \Gamma, \epsilon\}$ are obtained via fits of precalculated Mann-model spectra to the measured velocity-component and stress spectra $F_{11}(f)$, $F_{22}(f)$, $F_{33}(f)$, and $F_{13}(f)$; this is done via Taylor's hypothesis ($k_1 = 2\pi f/U$) and chi-squared fits (Mann, 1994; Chougule et al., 2017).

### 3.1  Testing of assumptions and predicted constraints

The implications of (12–15) included the independence of $L_{\mathrm{MM}}$ and $c_\tau$ from $\Gamma$, as well as e.g. the expected dependence $\sigma_{u,\mathrm{obs}} \propto \Gamma\sigma_{\mathrm{iso}}$. Indeed we find that $L_{\mathrm{MM}}$ is independent of $\Gamma$, with no significant statistical correlation: $\langle L_{\mathrm{MM}}\Gamma\rangle/\sqrt{\langle L_{\mathrm{MM}}^2\rangle\langle\Gamma^2\rangle} < 0.15$ for land or sea sectors at any given height. We also confirm that $\sigma_{u,\mathrm{obs}} \propto \Gamma\sigma_{\mathrm{iso}}$, which is demonstrated by Figures 1–2. The first figure displays the joint probability density $P(\sigma_{u,\mathrm{obs}}, \sigma_{\mathrm{iso}})$, where $\sigma_{u,\mathrm{obs}}$ is the streamwise turbulent variance measured in
10-minute intervals, and $\sigma_{\mathrm{iso}}$ is calculated using (9) with $L_{\mathrm{MM}}$ and $\epsilon$ from spectral fits corresponding to the same intervals. One can see from Fig. 1 that $\sigma_{u,\mathrm{obs}}$ generally follows $\sigma_{\mathrm{iso}}$, and we find $\sigma_{u,\mathrm{obs}}^2 \approx 3\sigma_{\mathrm{iso}}^2$; that is $\sigma_{u,\mathrm{obs}}/\sigma_{\mathrm{iso}} \approx 1.7$. Such evidence corresponds closely to the predicted constraint following (18) that $\sigma_{u,\mathrm{obs}}/\sigma_{\mathrm{iso}}$ should have a value of roughly 1.6 in the neutral surface layer; this is reasonable in the mean, since conditions on average are essentially neutral due to the shape of the stability distribution at Høvsøre (Kelly and Gryning, 2010). Figure 2 further shows that $\sigma_{u,\mathrm{obs}}/\sigma_{\mathrm{iso}} \propto \Gamma$, consistent with $c_\tau$ being a
constant independent of $\Gamma$ following (14). The slope of the line in Fig. 2 also corresponds to the behavior implied by the approximate Mann-model behavior $\sigma_{u,\mathrm{MM}} \approx \sigma_{\mathrm{iso}}(0.61 + 0.3\Gamma)$ for $\Gamma \gtrsim 2$, outlined at the end of section 2.2.2 above.

---

[6]The sonic anemometers actually give a temperature very close to the virtual temperature, i.e. the temperature including buoyant effects of water vapor.

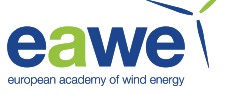
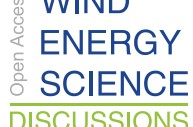




**Figure 1.** Joint distribution of isotropic (un-distorted) variance $\sigma^2_{\mathrm{iso}}(\varepsilon, L_{\mathrm{MM}})$ obtained from fits to measured spectra and observed streamwise variance $\sigma_{u,\mathrm{obs}}$, from height $z = 80\,\mathrm{m}$ over homogeneous land sectors at Høvsøre; dashed line indicates a slope of 3.



Considering only wind speeds above $7\,\mathrm{m\,s^{-1}}$ (i.e. ignoring low speeds that have little impact on turbine loads), the Høvsøre data also confirm that $\langle \sigma_u/u_* \rangle_{\mathrm{obs}} \approx 2.3$, consistent with the findings of Caughey et al. (1979). Further, for these significant wind speeds, the data also show that $\langle c_m \rangle \approx 2.3$, so that (13) reduces approximately to (15). Figures 1–2 were made considering $U > 7\,\mathrm{m\,s^{-1}}$, though they are essentially the same when including speeds down to $4\,\mathrm{m\,s^{-1}}$ (with slightly more scatter away

from the main trends shown). For the remainder of the figures, we continue to consider $U > 7\,\mathrm{m\,s^{-1}}$.

The data also show that $\sigma_u/u_*$ is not correlated with $L_{\mathrm{MM}}$, whether we include all speeds, or limit the wind speed range to $7$–$25\,\mathrm{m\,s^{-1}}$ or $4$–$25\,\mathrm{m\,s^{-1}}$. Thus this ratio can be treated as a constant in (13) for a given height (or throughout the surface layer), using (13) over a range of wind speeds.

### 3.2   Turbulence length-scale distributions $P(L_{\mathrm{MM}})$

The efficacy of using (15) to estimate the spectral length scale $L_{\mathrm{MM}}$ can be seen by considering Figure 3. The figure displays the joint-distribution of turbulence length scale at a height of $z = 80\,\mathrm{m}$, i.e. $P\left(L_{\mathrm{MM,obs}}, \sigma_{u,\mathrm{obs}}|dU/dz|^{-1}\right)$; this is obtained through (15) from 10-minute measurements and via fitting observed spectra. Fig. 3 is usefully interpreted as the probability-weighted performance of (15) for predicting $L_{\mathrm{MM}}$ (from $\sigma_{u,\mathrm{obs}}$ measured at $z = 80\,\mathrm{m}$ and the shear $dU/dz$ observed over $z = 60$–$100\,\mathrm{m}$), versus the $L_{\mathrm{MM}}$ obtained from fits of the spectral-tensor model to corresponding 10-minute spectra. One sees

a 1:1 relationship, particularly for the most commonly-found values of the length scale; these $L_{\mathrm{MM}}$ range from $\sim$15–50 m.[7] Compared to the scales calculated from observed spectra, there is some mis-prediction of $L_{\mathrm{MM}}$ calculated by (15), but it is relatively rare; this is shown by the low probabilities in Fig. 3 away from the well-predicted most commonly-ocurring $L_{\mathrm{MM}}$.

To demonstrate the statistical character of (15), as well as its potential for probabilistic use (e.g. as input to probabilistic loads calculations), Figure 4 shows the probability density $P(L_{\mathrm{MM}})$. As in Fig. 3, $L_{\mathrm{MM}}$ is again calculated from fits to 10-

minute spectra and also estimated by $\sigma_{u,\mathrm{obs}}/(dU/dz)$, i.e. Eq. 15. Additionally Fig. 4 displays $P(L_{\mathrm{MM}})$ for $L_{\mathrm{MM}}$ calculated through (11), i.e. $c_m u_*/(dU/dz)$; this is done both using the value of $c_m = 1.7$ reported by Peña Diaz et al. (2010), as well as using the approximate mean of 2.3 found to be consistent with measurements and theory in sections 3.1 and 2.2 above. From Fig. 4 one sees that for values of turbulent peak scale greater than the mode ($\sim$20 m) up to roughly 150 m, there is a match between the distribution of the diagnosed $L_{\mathrm{MM}}$ and distributions of length scale estimated from the forms (15) based on $\sigma_{u,\mathrm{obs}}$

and (11) based on $u_*$ with $c_m = 2.3$; these are roughly equivalent for this case over relatively simple homogeneous terrain. It is found that the Peña Diaz et al. (2010) value of $c_m = 1.7$ leads to overprediction of $L_{\mathrm{MM}}$ by a factor of 2 or more at scales smaller than 10 m, and underprediction by 50% or more at scales larger than 50 m. The $u_*$-based form (11) using $c_m = 2.3$ matches the spectrally-fit diagnosed distribution $P(L_{\mathrm{MM}})$ slightly better than the $\sigma_u$-based form (15), with predicted peak (mode) values of $L_{\mathrm{MM}}$ being about 3–4 m smaller than the diagnosed peak-$L_{\mathrm{MM}}$.

For the homogeneous land case in Fig. 4 the PDF of $2.3u_{*,\mathrm{obs}}/(dU/dz)$ matches $P(L_{\mathrm{MM}})$ observed from the spectral fits to within 10%, over the range $10\,\mathrm{m} \lesssim L_{\mathrm{MM}} \lesssim 75\,\mathrm{m}$, and the PDF of $\sigma_{u,\mathrm{obs}}/(dU/dz)$ also matches within 10% over the range

---

[7]The spectral fits were done using spectral-tensor model output over the parameter ranges of $5 < L_{\mathrm{MM}} < 500\,\mathrm{m}$ and $0 \leq \Gamma \leq 5$. Some spectra were poorly fitted; since these occurred when $\Gamma = 5$, cases with $\Gamma > 4.95$ were excluded from the analysis here. As justification, I note that only a small fraction of the cases ($< 10\%$) had such $\Gamma$, and that we only consider well-fit spectra for reliable comparison of parameters.





**Figure 2.** Ratio of observed streamwise to isotropic fluctuation magnitude, versus $\Gamma$ obtained from spectral fits, plotted as joint-PDF $P(\Gamma, \sigma_{u,\mathrm{obs}}/\sigma_{\mathrm{iso}})$. Dashed (horizontal) line shows $\sigma_{u,\mathrm{obs}}/\sigma_{\mathrm{iso}} = \sqrt{3}$ corresponding to slope of dashed line in Fig. 1; dotted line shows the mean linear $\Gamma$-dependence of $\sigma_{u,\mathrm{obs}}/\sigma_{\mathrm{iso}}$.


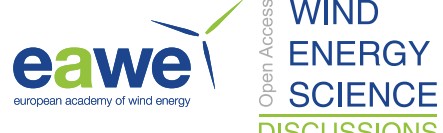

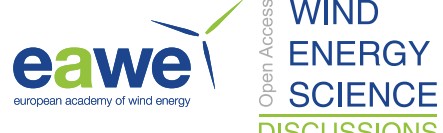

**Figure 3.** Joint probability density function of predicted and diagnosed (observed) turbulent length scale, from measurements at Høvsore over the homogeneous eastern land sectors. "$x$-axis": Mann-model scale $L_{\mathrm{MM}}$ from spectral fits; "$y$-axis": $L_{\mathrm{MM}}$ estimated from direct measurements of $dU/dz$ and $\sigma_u$, via (15).

$15\,\mathrm{m} \lesssim L_{\mathrm{MM}} \lesssim 50\,\mathrm{m}$. This is consistent with the darkly-colored 1:1 patch evident in Fig. 3, and also shows that eqn. 15 (and also eqn. 11 with $c_m = 2.3$) is sufficient for probabilistic wind loads simulations, for two reasons. First, the well-matched range of scales corresponds to the most commonly found $L_{\mathrm{MM}}$. Secondly, although scales smaller than ~15 m are not rare (with an occurence of roughly 1 in 6), they will have a diminishing effect on turbine loads. More specifically, $L_{\mathrm{MM}}$ is more than 70%




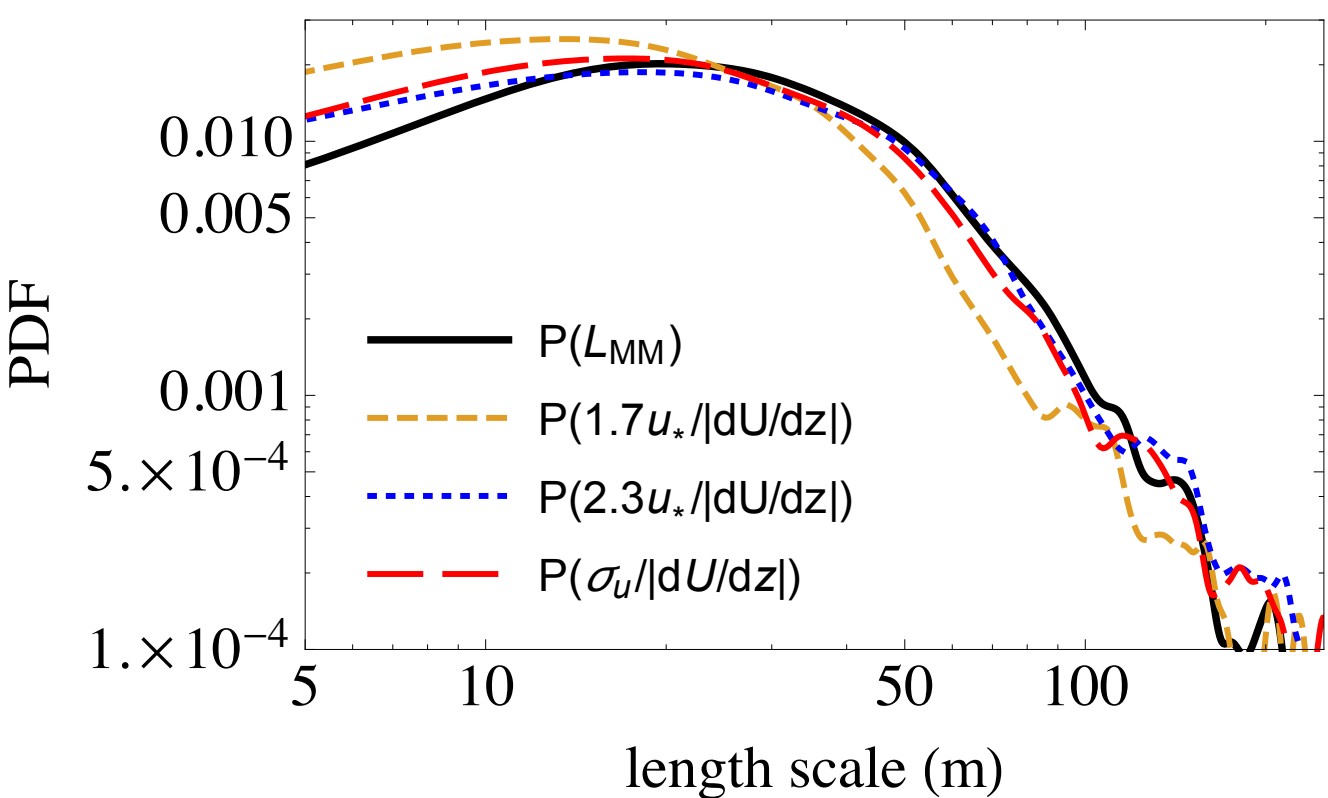

**Figure 4.** Probability density function of turbulent length scale from observations at Høvsore from the homogeneous eastern land sectors. Black: Mann-model scale from fits to spectra; dotted/blue: 'mixing-length' formulation ($\ell_m \propto u_*/|dU/dz|$) with revised constant; dashed/gold: Peña Diaz et al. (2010) form for $\ell_m$; red/long-dashed: $\sigma_u/|dU/dz|$ form (15).

likely to fall in the 15–75 m range, i.e. $P(15\,\text{m} < L_{\text{MM}} < 75\,\text{m}) > 0.7$, and $L_{\text{MM}}$ has more than 86% likelihood of occurence between 0 and 75 m, for this homogeneous land case at $z = 80\,\text{m}$. The relatively common shorter scales correspond to weaker turbulent fluctuations (thus loads), because on average $\sigma_{u,\text{obs}} \propto L_{\text{MM}}^{2/3}$ (as implied by Fig. 1 and Eqns. 9–15). Further, turbine loads are less influenced by fluctuations characterized by spatial scales significantly smaller than the blade lengths; thus the
5   error in predicted probability for these shorter scales, and the slight underprediction of the most common $L_{\text{MM}}$, should not significantly influence probabilistic loads calculations relying on site-specific $L_{\text{MM}}$ obtained via measurements and (15).

While (15) is useful to estimate $L_{\text{MM}}$ and $P(L_{\text{MM}})$ as shown above, one expects (13) to perform better, as it does not rely on the approximation $c_m = \sigma_u/u_*$. Indeed $\langle c_m u_*/\sigma_u \rangle$ is actually 1.13 (or 1.11 if considering winds down to $4\,\text{m s}^{-1}$) due to $\sigma_u/u_*$ being slightly smaller and $c_m$ slightly larger than 2.3; using these values in (13) gives estimates of $L_{\text{MM}}$ closer to the
10  spectrally-diagnosed $L_{\text{MM}}$, and within 10% of $P(L_{\text{MM}})$ over a range of $L_{\text{MM}}$ from below 10 m to beyond 100 m. It should also



be noted that also including speeds from $7\,\mathrm{m\,s^{-1}}$ down to $4\,\mathrm{m\,s^{-1}}$ can lead to slightly larger $L_{\mathrm{MM}}$, since these low wind speeds are more influenced by unstable conditions. Indeed for $L_{\mathrm{MM}} \gtrsim 50\,\mathrm{m}$, including the lower wind speeds caused both diagnosed and predicted $L_{\mathrm{MM}}$ to increase roughly 10%; this is consistent with larger turbulent eddies being created under unstable conditions.

### 3.2.1   Estimating $P(L_{\mathrm{MM}})$ in coastal/offshore conditions

To demonstrate the (probabilistic) use of (13) or (15) for $L_{\mathrm{MM}}$ in somewhat different conditions, we now consider flow from offshore, using data from the same mast and height as above (Høvsøre, $z = 80\,\mathrm{m}$) but for wind directions between $240°$ and $300°$. The mast is roughly $1.75\,\mathrm{km}$ east of the coastline and subsequently $1.65\,\mathrm{km}$ east of a $16$–$17\,\mathrm{m}$-high sand dune that lies $100\,\mathrm{m}$ inland, where both are locally oriented in the N-S direction (i.e. for the range of wind directions considered). The dune causes enhanced/accelerated transition of the flow from an offshore (water roughness) to an over-land flow-regime (Berg et al., 2015); this results in winds which reflect on-shore and coastal conditions at low heights (below $\sim$40–80 m depending on stability) and offshore conditions at higher $z$.

Figure 5 displays the distribution $P(L_{\mathrm{MM}})$ of spectral-peak (Mann-model) length scales for coastal/offshore winds (from west $\pm 30°$), again using (15) to estimate $L_{\mathrm{MM}}$ along with $L_{\mathrm{MM}}$ diagnosed through spectral fits. For comparison the corresponding $P(L_{\mathrm{MM}})$ for easterly winds from Fig. 4 is also included. Just as for the homogeneous land case shown in Fig. 4, one sees in Fig. 5 that for inhomogeneous coastal conditions, again (15) gives $P(L_{\mathrm{MM}})$ basically matching the spectrally-fit obervations for scales beyond $\sim$15 m; in this coastal regime the range of well-predicted $L_{\mathrm{MM}}$ extends further, to $\sim$150 m. While one sees that the distribution of $L_{\mathrm{MM}}$ is a bit different for the (western) inhomogeneous coastal case than for the (eastern) homogeneous land case, the simple expression (15) functions similarly for both flow regimes, with the arguments presented in above in section 3.2 again applying here. The $u_*$-based Eq. 11 also behaves similarly (not shown) as in the homogeneous land case of Fig. 4, i.e. with gross overpredictions at small scales and underpredictions at large scales. One difference between the coastal and land cases is that for small $L_{\mathrm{MM}}$, (15) overestimates the distribution $P(L_{\mathrm{MM}})$ a bit more for the coastal regime than for the homogeneous land regime ($L_{\mathrm{MM}} < 20\,\mathrm{m}$); as explained above for the land case, an overprediction at the smallest is not expected to significantly impact loads calculations, due to the relatively small length scales involved.

### 3.2.2   Estimation of $P(L_{\mathrm{MM}})$ in more complex conditions

To further show the behavior of $L_{\mathrm{MM}}$ and the utility of (15) at a site with more complex conditions, we examine data from the inhomogeneous forested Danish National Test Centre for Large Wind Turbines site near Østerild in Denmark (see e.g. Hansen et al., 2014, for details). Here sonic anemometer data is available at a heights of $10\,\mathrm{m}$ and $44\,\mathrm{m}$, with concurrent data from three lidars at $z = \{45, 80, 140, 200, 300\}\,\mathrm{m}$. In this study we consider data from the site's 'western LIDAR,'[8] to measure winds that flow over the forest more than 70% of the time, where the canopy height is 10–20 m (Hansen et al., 2014; Sogachev et al., 2017). The analysis here uses one year (May 2010–May 2011) of wind speeds $U \geq 5\,\mathrm{m\,s^{-1}}$ from the LIDAR at $45\,\mathrm{m}$ and $80\,\mathrm{m}$ heights along with the 'fast' (20 Hz) data from the sonic anemometer at $44\,\mathrm{m}$. The shear $dU/dz$ is measured across 45–80 m; the

---

[8]The 'western LIDAR' at Østerild is located $\sim$1 km west of the northern-most turbines but less than 100 m east of a forest patch and and 5–20 km from the North Sea coastline in the prevailing (W–NW) wind directions (Hansen et al., 2014).





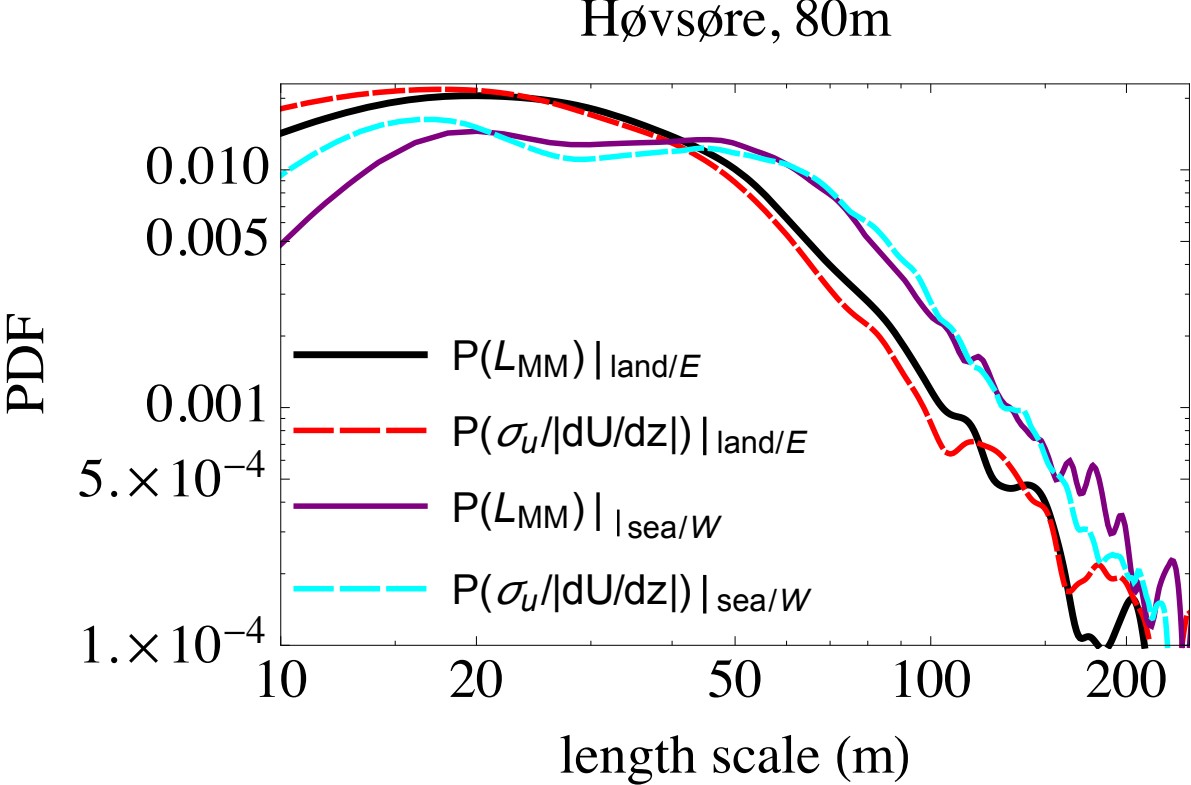

**Figure 5.** Probability density of turbulence length scale $L_{MM}$ from observations at Høvsore over both the homogeneous land (eastern) sectors and inhomogeneous coastal (western) sectors. Black: $L_{MM}$ from fits to spectra over land; red/long-dashed: new simplified form (15) over land; purple: $L_{MM}$ from fits to spectra from offshore; cyan/long-dashed: new simplified form (15) from offshore.

spectra and subsequent turbulence/Mann-model parameters $\{L_{MM}, \Gamma, \varepsilon\}$, as well as and measured quantities $\{\sigma_{u,obs}, u_{*,obs}\}$, are obtained from the sonic anemometer. The measurements are significantly higher than twice the forest canopy height, and thus above the roughness sublayer (Garratt, 1980; Raupach et al., 1980) and amenable to similarity and mixing-length theory (e.g. Sogachev and Kelly, 2016) as well as Mann-model use (Chougule et al., 2015).

5    Just as Figure 4 showed for flow over homogeneous land at Høvsøre in section 3, here Figure 6 displays the probability density of turbulence (Mann-model) length scale $L_{MM}$ observed via spectral fits at $z = 44$ m for Østerild, along with predictions based on both (11) via $u_{*,obs}$ and (15) via $\sigma_{u,obs}$. As in the cases above (homogeneous land and inhomogeneous coastal), the new form (15) predicts the distribution rather well, particularly for scales between $\sim$10–100 m—despite the shape of $P(L_{MM})$ being different due to the trees. For the forest case of Fig. 6 the $\sigma_u$-based form captures both the peak (most likely $L_{MM}$)

10   and magnitude of $P(L_{MM})$, while the $u_*$-based form grossly underpredicts $L_{MM}$, moreso than for the previous cases. The latter is likely due to $u_{*,obs}$ being affected by the canopy (via its larger effective roughness) moreso than $\sigma_{u,obs}$, which tends to be more characteristic of the entire ABL. There is, however a curious minor peak (with a probability well under 1%) for





around $\sim$300$\pm$50 m in the length-scale distribution obtained from spectral fits shown in Fig. 6, which is not captured by either formulation. Although this peak falls spectrally at small wavenumbers (frequency divided by mean wind speed) that are more difficult to capture when spectrally fitting the Mann model, it indeed corresponds to the distance to the next upwind edge of the forest (orchard segment) in the predominant wind directions.

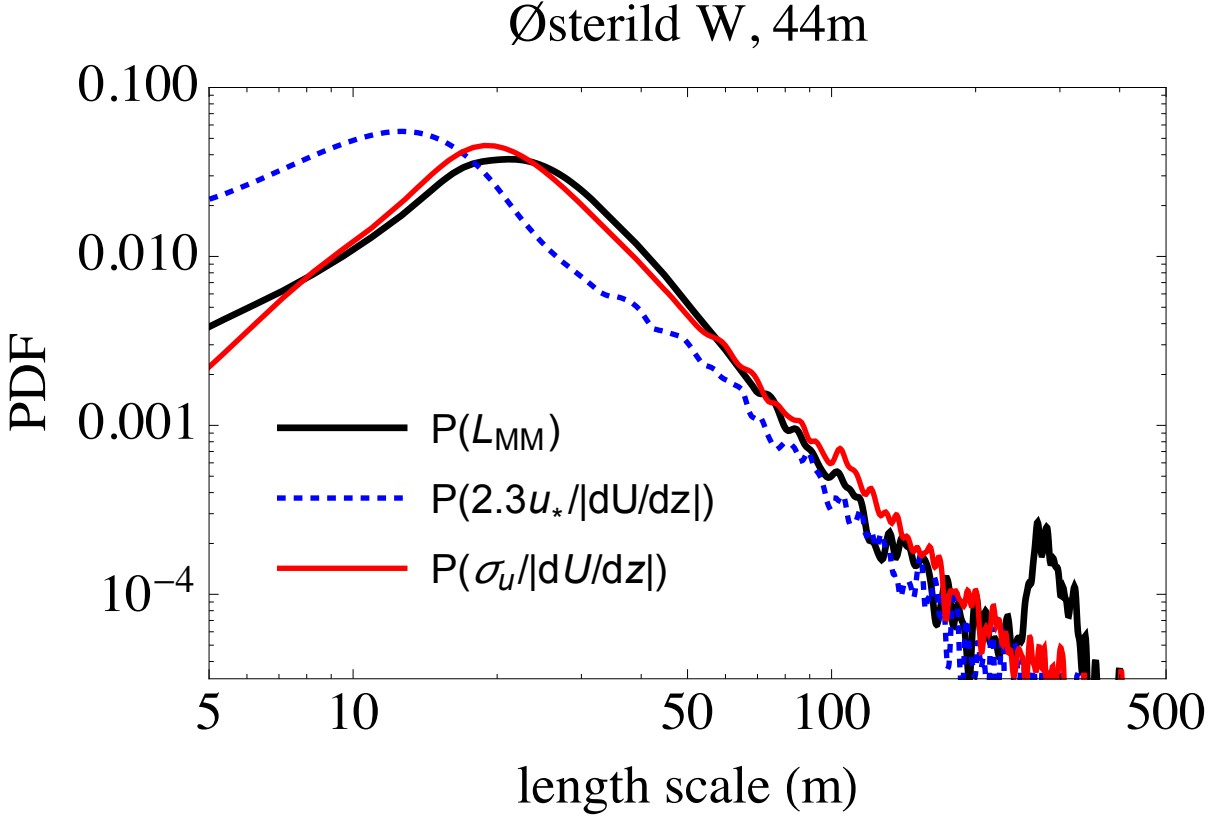

**Figure 6.** Probability density function of turbulent length scale from observations at Østerild from the western mast/lidar. Black: Mann-model scale from fits to spectra; dotted-blue: 'mixing-length' formulation ($\ell_m \propto u_*/|dU/dz|$) with revised constant; red: new form (15), $\sigma_u/|dU/dz|$.

## 4 Conclusions

Towards concluding, we first revisit the motivation for (and thus context of) this work: [1] to 'close' the Mann (1994) eddy-lifetime ($\tau_M$) formulation as implemented in rapid-distortion theory—allowing relation between Mann-model parameters ($L_{MM}, \varepsilon, \Gamma$) and the shear ($dU/dz$) taken to distort the modeled turbulence; [2] connect the parameters of the Mann (1994) spectral turbulence and eddy-lifetime models with atmospheric statistics, both in theory and in practice; [3] provide a formulation for the turbulence length scale $L_{MM}$ in terms of quantities commonly-measured in wind energy; [4] demonstrate

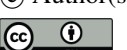



that the 'measureable' form developed for $L_{\mathrm{MM}}$ is robust and amenable to use in (probabilistic) wind turbine loads calculations. These four motivating goals have basically been realized, as shown in the previsous sections, and this work has a number of implications.

## 4.1  Implications and Application

A previously suggested form (11) for $L_{\mathrm{MM}}$, based on friction velocity $u_*$ and (10-minute) mean wind shear $dU/dz$ (Peña Diaz et al., 2010), was confirmed here to be sensitive to its proportionality constant $c_m$. But this constant can vary from site to site (and possibly with height), and the published value of $c_m = 1.7$ (Peña Diaz et al., 2010) leads to significant error in prediction of $L_{\mathrm{MM}}$ for the different conditions (land and sea directions) at Høvsøre and at the forested site of Østerild. Finding $c_m$ from sonic anemometer observations via $L_{\mathrm{MM}}$ from fits to spectra and friction velocity measurements, (11) may perform slightly better over uniform flat terrain compared to the $\sigma_u$-based form (15); but this can be considered a site-dependent fit in itself, as was the case when using a diagnosed value of $c_m = 2.3$ for the homogeneous flat land sectors at Høvsøre. However, obtaining $c_m$ is generally not possible in industrial practice; where it can be obtained, it relies on $L_{\mathrm{MM}}$—which is the quantity desired— thus negating the purpose of (11). While $u_*$ can also in principle be estimated from wind speeds taken at multiple heights by cup anemometers, this too is difficult in practice: one must account for stability, not to mention the need for measurements at multiple heights in the surface layer (or worse, the limited validity of similarity theory above the ASL). Furthermore, it is expected that $c_m$ is a function of the (local) surface roughness, as demonstrated by the different results found over the forested Østerild site. Thus the form (15) is preferable, since it requires only the commonly-measured quantities $\sigma_u$ and $dU/dz$. This simple form also gave good estimates of $P(L_{\mathrm{MM}})$ in the forested case—without the need for tuning, whereas the $u_*$-based form (11) requires a re-calculation of its coefficient $c_m$ for such cases.

Since (13) gave yet better performance than both its simplified form (15) and the $u_*$-based relation (11), one might suggest its use. But (13) requires $c_m/(\sigma_u/u_*)$, where $c_m$ is difficult to obtain, as discussed in the previous paragraph. However, although $c_m$ might vary from site to site (or perhaps with height), it was found that the ratio $c_m/(\sigma_u/u_*)$ did *not vary* appreciably— consistent with the good performance of the simplified form (15), which assumes $c_m \approx \sigma_u/u_*$, across sites and regimes.

One interesting implication of the testing of assumptions then follows from the finding that $\langle \sigma_u/u_* \rangle_{\mathrm{obs}} \approx 2.3$, consistent in the surface-layer with Caughey et al. (1979). Examining the joint behavior of $\sigma_u/u_*$ and the stability parameter (inverse Obukhov length) $L^{-1}$, the sonic anemometer data available at multiple heights in this study shows no correlation between these two quantities. The dimensionless profiles $\sigma_u^2(z)/u_*^2$ and $u_*^2(z)/u_{*0}^2$ of Caughey et al. (1979) also imply

$$\frac{\sigma_u^2(z)}{u_*^2(z)} \approx (2.3)^2, \tag{19}$$

with the ratio particularly converging to a constant above the surface layer ($z/h \gtrsim 0.1$, where $h$ is the atmospheric boundary-layer depth). The flat-terrain Høvsøre data in fact show the mean value $\langle \sigma_u/u_* \rangle_{\mathrm{obs}}$ to be independent of $z$. If one knew the height-dependent behavior of $c_m$, then one could also use (19) and (13) from measurements at one height range, to estimate $L_{\mathrm{MM}}$ at higher $z$. Since the the peak spectral scale for streamwise fluctuations ($\lambda_u$) grows with $z$ (Caughey et al., 1979; Peltier





et al., 1996)[9], if we take $L_{\mathrm{MM}} \propto \lambda_u$ then with (19) one expects the ratio $c_m/(\sigma_u/u_*)$ to increase with $z$ as well. Thus from (13) the Mann-model length scale $L_{\mathrm{MM}}$ will increase with height relative to the mixing-length $\ell_* \equiv u_*/(dU/dz)$, so at higher $z$ one would expect the general form (13) to be yet more accurate than its approximate form (15); though this is not likely for wind turbine rotor heights, except in very stable conditions (Kelly et al., 2014b; Liu and Liang, 2010). Unfortunately the sonic-anemometer measurements available for this study did not include heights well beyond the surface-layer, so such variation was difficult to detect.

It is also notable that Figure 3 appears to imply the relative error (e.g. in %) in estimating $L_{\mathrm{MM}}$ with (15) grows for less common values of $L_{\mathrm{MM}}$, particularly very large scales (and also at very small scales if including $U < 7$m/s, not shown). Thus (15) is recommended first for estimation of $P(L_{\mathrm{MM}})$. However, the error at large scales is in part dependent on the limited (10-minute) sample lengths and the fitting routine, as there are very few points to fit at the lowest frequencies. Use of 30-minute samples can reduce such scatter, and modification of the fitting algorithms may also improve estimations of the larger scales.

Ongoing work includes wind-speed dependent prediction of $L_{\mathrm{MM}}$, particularly the conditional statistics $P(L_{\mathrm{MM}}|U)$. Further concurrent work also entails systematic accounting for the rotor size (shear distance) relative to height (i.e. $\Delta z/z$) within the distribution of length scales; following Kelly and Gryning (2010) and Kelly et al. (2014a) a semi-empirical derivation of $P(L_{\mathrm{MM}})$ including $\Delta z/z$ has been obtained, but demands more data for validation and publication. Understanding of the latter facilitates 'vertical extrapolation' of $L_{\mathrm{MM}}$ and measured turbulence and shear statistics, as well as accounting for the effect of rotor size or shear measurement span.

## 4.2 Summary of conclusions

- The eddy lifetime of Mann (1994), which is part of commonly used turbulence modelling for wind turbine design load cases (e.g. IEC 61400–1, Edition 3, 2005), leads to a relation for turbulence (spectral-peak) length scale $L_{\mathrm{MM}}$ of

$$L_{\mathrm{MM}} \simeq \frac{c_m}{(\sigma_{\mathrm{u,obs}}/u_{*,\mathrm{obs}})} \frac{\sigma_u}{dU/dz},$$

  where $c_m$ and $\sigma_{\mathrm{u,obs}}/u_{*,\mathrm{obs}}$ are essentially constants for a given height $z$.

- Theory and measurements support the assumption that $c_m/(\sigma_{\mathrm{u,obs}}/u_{*,\mathrm{obs}}) \approx 1$, roughly constant for different atmospheric flow regimes; the turbulence length scale can thus be approximated as

$$L_{\mathrm{MM}} \simeq \frac{\sigma_u}{dU/dz};$$

  thus typical 10-minute mean cup anemometer measurements can be used to estimate $L_{\mathrm{MM}}$.

- $L_{\mathrm{MM}}$ is affected by atmospheric stability; this effect is contained within $\sigma_u$ and $dU/dz$.

---

[9]The peak length scale also grows with boundary-layer depth $h$ in convective conditions and in general with decreasing (increasingly negative) inverse Obukhov length $L^{-1}$ (e.g. Peltier et al., 1996). But over all stability conditions, which are dominated by neutral conditions (Kelly and Gryning, 2010), and over an expected distribution of $h$ at a given site, the basic growth of $\lambda_u$ with $z$ is consistent with Peltier et al. (1996) reporting $\lambda_u \propto z$ for neutral conditions.



- In terms of the classic mixing-length form $u_*/|dU/dz|$, the turbulence length scale $L_{\mathrm{MM}}$ in the spectral-tensor model is observed to be larger (by ca. 30–40%) than previously reported by Peña Diaz et al. (2010).

*Acknowledgements.* The author thanks the reviewers for their time and effort towards constructive criticism of the present article; and thanks to Nikolay Dimitrov for discussions around probabilistic loads. This work was partly supported by the DTU Wind internally-funded cross-
5 sectional project "*Wind to Loads*".



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
