# Peer review of "From standard wind measurements to spectral characterization: turbulence length scale and distribution"

_Wind Energy Science, 2018_

## Author Comment (AC1) · 19 Mar 2018

The online 'Discussion' paper included an old, incomplete version of the introduction. The complete introduction is attached here (including missing line from beginning of section 2), as a supplement.

The file will have a name like "wes-2018-14-supplement.pdf ".

I apologize for the inconvenience.

Please also note the supplement to this comment:

https://www.wind-energ-sci-discuss.net/wes-2018-14/wes-2018-14-AC1-supplement.pdf

[Figure]

**Supplement:**

**1 Introduction**

[revised manuscript text omitted]

---

## Referee Comment (RC1) · Anonymous Referee #1 · 3 Apr 2018

Summary

This paper presents a derivation of the turbulent length scale as a function of standard deviation and wind profile using the Mann model (Mann, 1994) and also closely following the derivations provided in de Maré and Mann (2016). This reviewer believes that the manuscript has potential to be published, but first several clarifications are needed. Please see the full list of my comments below.

Comments

1. After Eq. (3) define $L_{MM}$ as the turbulent length scale in Mann-model. You described all other parameters except for $L_{MM}$.
2. Although this reviewer is not a native English speaker, I would suggest that the authors uses less parentheses and footnotes if possible. For instance, the last sentence in Section 2.1.1 (around Line 20 on Page 4) contains many commas and a semicolon and parentheses that makes it difficult to understand. Similar examples can be found elsewhere in the manuscript.
3. Sometimes you are using Figure and sometimes Fig. for figures (in Section 3.1 and later). Please be consistent.
4. Font size in your figures is very large. I am not sure if this will be handled in the production stages if the manuscript gets accepted, but if not, you should decrease the font size.
5. Please specify the frequency of the occurrence of wind speeds above 7 m s$^{-1}$ at the Høvsøre mast. Why 7 m s$^{-1}$ and not, for instance, 5 m s$^{-1}$?
6. Section 4.1 (Implications and Applications) should not be a part of the concluding section. Conclusions should conclude the study and not elaborate on the applications of the result. Please move Implications and Applications prior to Conclusions and remove the subsection title Summary of conclusions (Section 4.2). It is not typical to have subsections in conclusions.
7. To this reviewer, current Section 4.1 is a typical discussion section and not implications and applications. I suggest the author renames this section to discussion.
8. The author concludes (e.g., Line 21 on Page 17) that $L_{MM}$ is influenced by atmospheric stability but the analyses in this study are not conducted for unstable, stable and neutral conditions separately. Nothing has been said about the fluxes of sensible heat, Richardson number, Obukhov length, etc.
9. Related to my previous comment, the paper by Peña Diaz et al. (2010) clearly lists the stability classes that were investigated (Table I in that article), so it would be useful to see similar analysis in this paper.
10. Please clarify the purpose of Section 2.2.2 (Modelled spectra: Covariances, anisotropy and Γ) and Section 2.3 (Ideal, neutral and surface-layer implications). All figures referee to Eq. (15) and the expressions prior to that equation. I don't see how these sections contribute to the manuscript.
11. Please discuss the reasons why the peak in the Mann model in Figure 6 is not captured by the other two models? This peak, although at small wavenumbers, is very prominent and should be explained. Please discuss.
12. What is the sampling frequency of the lidar data? The peak in Figure 6 seem not to appear in Figure 5, so is it possible that the lidar measurements contain some bias or some filtering was applied (or something else)?
13. Please specify the source for Eq. (6).
14. I recommend that the author writes the alternative equation for $L_{MM}$ in Line 27, Page 5 as a numerated equation and not an in-line expression [i.e., Eq. (16)] since some researchers might

prefer the usage of turbulence intensity and shear exponent over standard deviation and wind profile (or maybe they already have the data in the form of $I$ and $\alpha$).

---

## Short Comment (SC1) · 17 May 2018

Dear Mark,

Thanks for a very interesting paper. It is indeed extremely convenient to have a parametrization for the Mann length scale that is based on commonly measured parameters. Here three short comments on your manuscript:

1. My previous work both in the citations and in the references should be Peña, A and not Peña Diaz, A. I think you have two references (and the corresponding citations) with that issue.

2. In Peña et al. (2010) we did not explicitly suggest a parametrization for the Mann length scale but we relate it to the length scale of the wind profile as you point out. Your work suggests $L_{MM} \approx \sigma_U/dU/dz$ which roughly means that $L_{MM} \approx z$ in the surface layer (if the approximation $\sigma_U = u_*/\kappa$ is used), whereas our relation $L_{MM} \approx 1.7\ell$ roughly means $L_{MM} \approx 0.68z$. The latter is also in accordance with the work of Chougule et al. (2014) from measurements at Høvsøre and at Ryningsnäs.

3. So what is the reason for the differences between Peña et al. (2010)/Chougule et al. (2014) and your results? Could it be the way the velocity spectra was analyzed (you seem to extract the Mann parameters from each individual 10-min record whereas Peña et al. (2010)/Chougule et al. (2014) ensemble average spectra for different turbulence conditions)? What is the uncertainty of the fit when performed on each 10-min case?

Best regards,

Alfredo
* * *

---

## Referee Comment (RC2) · Anonymous Referee #2 · 22 Jun 2018

[referee-annotated manuscript omitted]

---

## Author Response (AR1)

**Author's reply (AC2) to reviewer #1's comments (RC1)**

First I (the author) would like to thank the anonymous reviewer (#1) for constructive criticism, towards improvement and clear dissemination.

I'll include the reviewer's comments as I respond point-by point, starting with the summary and proceeding through the comments.
The reviewer's comments are indented/italic, including their original numbers.

> *"Summary:*
> *This paper presents a derivation of the turbulent length scale as a function of standard deviation and wind profile using the Mann model (Mann, 1994) and also closely following the derivations provided in de Mare and Mann (2016). This reviewer believes that the manuscript has potential to be published, but first several clarifications are needed. Please see the full list of my comments below."*

I would argue that the derivations in this paper **do not** 'follow' those in de Mare & Mann (2016). While the new expression (5) can be compared to an analogous one in de Mare & Mann, most of the expressions I derived don't have any correspondence or equivalent in de Mare & Mann—e.g. the simple practical (and perhaps most important) expression $L_{MM} \simeq \sigma_u/(dU/dz)$. I should add that the derivations in this work were done in 2015–16 (except the new generic vonKarman-simplification in eq.7); i.e. the work was done independently and concurrently in a different project than de Mare & Mann (2016).

> 1. *"After Eq. (3) define $L_{MM}$ as the turbulent length scale in Mann-model. You described all other parameters except for $L_{MM}$."*

This error due to editing is now corrected in the revision.

> 2. *"Although this reviewer is not a native English speaker, I would suggest that the authors uses less parentheses and footnotes if possible. For instance, the last sentence in Section 2.1.1 (around Line 20 on Page 4) contains many commas and a semicolon and parentheses that makes it difficult to understand. Similar examples can be found elsewhere in the manuscript."*

I have attempted to use footnotes in such a way as to preserve the flow of the main text, so that details are available to the interested reader while minimally interrupting the flow.

However, as reviewer #1 points out, there are some relatively convoluted sentences. I have worked to clean up/clarify these in the revision.

3. *"Sometimes you are using Figure and sometimes Fig. for figures (in Section 3.1 and later). Please be consistent."*

I intentionally use 'Fig.' in some passages to avoid overusing the word 'Figure' in places where more references to figures occur. Checking the WES manuscript guidelines, this appears to be ok (I'd prefer to leave it, unless WES objects per their English style preferences).

4. *"Font size in your figures is very large. I am not sure if this will be handled in the production stages if the manuscript gets accepted, but if not, you should decrease the font size."*

Such 'big' figures were made for scaling to 1-column width (half of current size) in the final publication.

5. *"Please specify the frequency of the occurrence of wind speeds above $7\,m\,s^{-1}$ at the Høvsøre mast. Why $7\,m\,s^{-1}$ and not, for instance, $5\,m\,s^{-1}$?"*

As written/mentioned, this was done with consideration of loads in a concurrent project—given the relatively infrequent occurence, lower impact on loads, higher difficulty fitting spectra in that regime, and larger spread of results.

Considering winds above cut-in, $P(U > 7\,\mathrm{m\,s^{-1}})$ for the period analyzed is 66% for the land case and 81% for the offshore case.

I updated the analysis to be for the range 4–25 m/s (where $U > 7$m/s is noted for its slight difference) and re-made the plots.

The conditional dependence of $L_{MM}$ on wind speed is beyond the scope of this paper, but is the subject of ongoing work.

6. *"Section 4.1 (Implications and Applications) should not be a part of the concluding section. Conclusions should conclude the study and not elaborate on the applications of the result. Please move Implications and Applications prior to Conclusions and remove the subsection title Summary of conclusions (Section 4.2). It is not typical to have subsections in conclusions."*

I revise based on your suggestions.

7. *"To this reviewer, current Section 4.1 is a typical discussion section and not implications and applications. I suggest the author renames this section to discussion."*

I updated to make this part of the discussion section.

> 8. *"The author concludes (e.g., Line 21 on Page 17) that $L_{MM}$ is influenced by atmospheric stability but the analyses in this study are not conducted for unstable, stable and neutral conditions separately. Nothing has been said about the fluxes of sensible heat, Richardson number, Obukhov length, etc.*
> 9. *Related to my previous comment, the paper by Pena Diaz et al. (2010) clearly lists the stability classes that were investigated (Table I in that article), so it would be useful to see similar analysis in this paper."*

Responding to points 8–9 together: explicit stability considerations are beyond the scope of the current article. Part of the point of this paper is that for application to loads, where one is concerned most with $\{\sigma_u, U, \alpha, L_{MM}\}$, which are affected by stability, one then needs to get $L_{MM}$ (the other 3 are easily obtained). We are not concerned here with stability itself—as the turbines are *not directly affected* by stability, as (re-)stated in the article and references cited.

However, in parallel work (in preparation for publication) and in related recent articles with Chougule et al (cited) we/I have examined treatment of stability.

Again, this is the subject of another paper, particularly because stability does not have a direct affect—but acts through $\sigma_u, U, \alpha(dU/dz)$ and $L_{MM}$.

> 10. *"Please clarify the purpose of Section 2.2.2 (Modelled spectra: Covariances, anisotropy and $\Gamma$) and Section 2.3 (Ideal, neutral and surface-layer implications). All figures referee [sic] to Eq. (15) and the expressions prior to that equation. I dont see how these sections contribute to the manuscript."*

Section 2.2.2 shows the theoretical self-consistency of the derived $\tau_M$ and model, with regard to $u_*$ (i.e. shear stress) and $\sigma_u$ and w.r.t. the mixing-length relation. Along the way, §2.2.2 also gives practical/understandable expressions for how Mann-model $\sigma_u$ and $u_*$ depend on $\Gamma$.

Section 2.3 shows the surface-layer limit of the derived $L_{MM}$; previously it was assumed that the Mann-model is basically designed to work in this limit. Further, §2.3 derives the expected asymptotic (neutral/equilibrium) relation connecting observed $\sigma_u$ and the model-constraining $\sigma_{iso}$.

> 11. *"Please discuss the reasons why the peak in the Mann model in Figure 6 is not captured by the other two models? This peak, although at small wavenumbers, is very prominent and should be explained.*

*Please discuss."*

As discussed in the text, this minor peak is not prominent ('probability well under 1%'). Note Fig. 6 is plotted in log-log coordinates, and these larger $L_{MM}$ in the minor bump are less than 1/1000 times likely than the values occurring around the major peak. I should adjust 'under 1%' to become 'under 0.1%'.

This minor peak is likely not captured because information related to its cause is not carried through $dU/dz$, but rather within horizontal gradients—which implicitly affect the fit parameters including $L_{MM}$. I did not wish to speculate, without more detailed measurements; this kind of advective artifact is not trivial do pick apart, given the conditions and the difficulty of fitting spectra to the Mann-model when the observed spectral-peaks are at smaller wavenumbers.

12. *"What is the sampling frequency of the lidar data? The peak in Figure 6 seem not to appear in Figure 5, so is it possible that the lidar measurements contain some bias or some filtering was applied (or something else)?"*

As mentioned above this peak is rather rare and corresponds to the distances to a forest edge. The LIDAR are not the cause, as the peak comes from the sonic anemometer; using data from the sonics only (over smaller vertical extent), the same trend (no peak) arises as when using the LIDAR. Further, the LIDAR and sonic data at $45/44\,\mathrm{m}$ are giving values almost identical to each other.

13. *"Please specify the source for Eq. (6)."*

Equation 6 follows from $\tau_M$ integrated explicitly using the vonKarman spectrum: eq.3 is equal to eq.5. I now add mention of (3) and (5) being equal in the text preceeding (6), to avoid confusion.

14. *"I recommend that the author writes the alternative equation for $L_{MM}$ in Line 27, Page 5 as a numerated equation and not an inline expression [i.e., Eq. (16)] since some researchers might prefer the usage of turbulence intensity and shear exponent over standard deviation and wind profile (or maybe they already have the data in the form of I and $\alpha$)."*

Amusingly in an earlier working draft I had actually done this, but removed it, thinking I had too many equations. But I agree and will switch back to having a separate numbered equation for $L_{MM} \simeq zI_{obs}/\alpha$.

**Author's reply (AC4) to reviewer #2's comments (RC2)**

The author (MK) would like to thank reviewer #2 for the compliments and constructive suggestions.

Here I will respond to the points raised by the reviewer, copying their points (from their annotation of the draft manuscript) inside quotes using *italic font*, and including page/line numbers:

1. p.1, lines 14–17 *"This looks very useful during the design phase of a wind-farm, particularly offshore."*

   Thanks; I hope it's useful, and look forward to get more offshore measurements, at 'taller' heights, to further verify the model—as I extend it conditionally per wind speed.

2. p.2, lines 6–8 *"While $L_{MM}$ is certainly one of the central Mann model parameter, anisotropy parameter $\Gamma$ is also quite important. In the IEC standard, it is recommended to use $\Gamma = 3.9$, but its value also varies under different stability conditions. Therefore, I suggest to tone down the 'the most relevant' to 'critical', so that $\Gamma$ is not forgotten :-) "*

   As mentioned and referenced in the text, Dimitrov and others found that $L_{\mathrm{MM}}$ is more relevant than $\Gamma$ for modern horizontal-axis turbines (and control systems) analyzed; e.g. Sobol coefficients for $\Gamma$ have been found to be much smaller than those for $L_{\mathrm{MM}}$. But there is a (small) possibility that in some circumstance (turbine and/or control system configuration) for some component load that the sensitivity to $\Gamma$ could be higher than for the turbulence length scale. The variation in $\Gamma$ is also mentioned, to avoid 'forgetting' it as well—the text reads "most relevant load–driving parameters", and this includes $\Gamma$.
   But I change 'relevant' to 'crucial,' inspired by the reviewer's suggestion.

3. p.2, line 20 (equation 1) *"Please add a reference to this equation."*

   There is no reference for this equation; rather it is a generic finding of the author, which corresponds to/relates all of the different forms of $\tau$ found in the literature (and referenced). (Such an expression could be useful in the future for e.g. fractal turbulence considerations.)

4. Figures 1–2 (p.8,10) *"Please add a legend indicating magnitude of joint probabilities, which I guess is hidden in the color intensity."*
   Done.

5. p.12, lines 7–10 *"Is Eq. (13) then recommended to use instead of Eq. (15), by using the ratio in the bracket to be 1.11/1.13?"*

   The value of 1.11 (or 1.13) corresponds to deviation from $\langle c_m u_*/\sigma_u \rangle = 1$

for an average including all recorded speeds between 4–25 m/s (or 7–25 m/s). If one wished to consider speeds only above 7 m/s at this site, then once could perhaps approximate the growth of this factor by the ratio 1.13/1.11—but this is found thus far only for this site and wind speed ranges. Later text (following this sentence) explains more about $\langle c_m u_*/\sigma_u \rangle$.

6. p.17, line 19 (second bullet-point in summary of conclusions/§4.2) *"On page 12 in the last paragraph, it seems that argument is made in favour of the ratio >1. Therefore, I suggest clarifying this in relation to those statements."*

   Note the ratio is '≈1' in the statement/second bullet point; the statement goes on to say that $L_{\mathrm{MM}}$ can then be *approximated* by $\sigma_u/(dU/dz)$. I have added a sentence to the end of the previous bullet-point, noting that this ratio can be 1–1.11 (or re-directing a reader of only the conclusion to check out the details).

**Author's reply (AC3) to comments by A. Peña (SC1)**

*"Thanks for a very interesting paper. It is indeed extremely convenient to have a parametrization for the Mann length scale that is based on commonly measured parameters. Here three short comments on your manuscript:"*

Thanks; I'm hoping to provide something which is theoretically and empirically sound, and convenient to use in wind applications.

*"1. My previous work both in the citations and in the references should be Peña, A and not Peña Diaz, A. I think you have two references (and the corresponding citations) with that issue."*

Ok, I'll update my BibTeX entries that include your name.

*"2. In Peña et al. (2010) we did not explicitly suggest a parametrization for the Mann length scale but we relate it to the length scale of the wind profile as you point out. Your work suggests $L_{MM} \approx \sigma_u/(dU/dz)$ which roughly means that $L_{MM} \approx z$ in the surface layer (if the approximation $\sigma_u \approx u_*/\kappa$ is used), whereas our relation $L_{MM} \approx 1.7\ell$ roughly means $L_{MM} \approx 0.68z$. The latter is also in accordance with*

> *the work of Chougule et al. (2014) from measurements at Høvsøre and at Ryningsnäs."*

First, this is only approached in the *neutral* surface layer (ASL).
Secondly, for $\sigma_u/u_* \approx 2.3$ (as shown in sections 2–3, and also found for the data sets in the neutral ASL), then $L_{MM}|_{\mathrm{nASL}} \approx 2.3z/\kappa \simeq 0.92z$ as given at the beginning of section 2.3.

Chougule *et al.* (2014, e.g. Fig. 5) actually shows agreement with $L_{MM} \sim z$ in the ASL ($z = 20$m) at Høvsøre (though their analysis is only for $U$ between 7–8 m/s). At Ryningsnäs, when accounting for the displacement height ($d \simeq 13$m) then their results are again consistent with the above, with $L_{MM} \approx z - d$ or actually slightly larger (though affected by roughness-sublayer effects above the forest there).

> *"3. So what is the reason for the differences between Peña et al. (2010)/Chougule et al. (2014) and your results? Could it be the way the velocity spectra was analyzed (you seem to extract the Mann parameters from each individual 10-min record whereas Peña et al. (2010)/Chougule et al. (2014) ensemble average spectra for different turbulence conditions)? What is the uncertainty of the fit when performed on each 10-min case?"*

As noted in my response to point 2 above, in the *neutral surface layer* there are not significant differences.

Overall, the increase of $L_{MM}$ in unstable conditions is significantly larger than the decrease in stable conditions, as also implied e.g. in Sathe *et al.* (2012). The vertical range and extent to which $\langle L_{MM}\rangle \sim z$ in all conditions depends on the (relative) widths of the stable- and unstable sides of the stability distribution $P(1/L)$ as well as the distribution of ASL depth.
As for the uncertainty on spectrally-fit $L_{MM}$, this is beyond the scope of the current article—though I do note that the fit was improved markedly by rejecting $\Gamma > 4.95$ (which corresponds to the fit using the highest $\Gamma$ of the lookup-table of Mann-model outputs), and such rejection roughly appeared to eliminate potential bias in $L_{MM}$; the latter is included as a footnote in section 3.2. Continuing work includes checking such fitting uncertainty/variability, as well as analysis per wind speed bin.

[revised manuscript text omitted]
_{\mathrm{vK}}(k)$ occurs at $kL_{\mathrm{MM}} = \sqrt{12/5}$, i.e. $L_{\mathrm{MM}} \simeq 1.55/k_{\mathrm{peak}}$.

[6] Note $\sqrt{2/3} \simeq 0.82$, and $3.07 = a_{HG}^{-3}$; c.f. footnote 4. In (6), $\alpha\varepsilon^{2/3}$ is kept together for comparison with (4), and because $\alpha\varepsilon^{2/3}$ is commonly used as an input to the spectral-tensor model instead of $\varepsilon$ (e.g. Mann et al., 2002; IEC 61400–1, Edition 3, 2005).

Since (3) and (5) are equal, we have an expression relating the Mann-model parameters to the shear $dU/dz$:

[revised manuscript text omitted]

$$
c_\tau \simeq \frac{1.5\Gamma}{c_m} \frac{\sigma_{\text{iso}}}{u_{*,\text{obs}}} = \frac{1.5\Gamma}{\sigma_{\text{u,obs}}/\sigma_{\text{iso}}} \left( \frac{\sigma_{\text{u,obs}}/u_{*,\text{obs}}}{c_m} \right).
\tag{12}
$$

From the above and (10) one subsequently then finds

$$
L_{\text{MM}} \simeq \frac{\sigma_{u,\text{obs}}}{dU/dz} \left( \frac{c_m}{\sigma_{\text{u,obs}}/u_{*,\text{obs}}} \right).
\tag{13}
$$

For constant $(\sigma_{\text{u,obs}}/u_{*,\text{obs}})$, (13) implies that the turbulence scale $L_{\text{MM}}$ can be expressed *independently* of $\Gamma$, given $\sigma_{u,\text{obs}}$ and $dU/dz$.

Caughey et al. (1979) reported the mean profile of $\sigma_u^2(z)$ from the seminal 'Kansas experiment', showing that $(\sigma_u/u_*)_0^2 \approx 5$–6 in the homogeneous atmospheric surface layer (their Fig. 5). The corresponding value of $(\sigma_u/u_*)_0$ is approximately 2.3; thus, if $c_m \approx 2.3$ as well, then (12) reduces to

$$c_\tau \approx \frac{1.5\Gamma}{\sigma_{u,\text{obs}}/\sigma_{\text{iso}}}. \tag{14}$$

Given the definition of $c_\tau$ through (7), $c_\tau$ is a constant; since (9) shows $\sigma_{\text{iso}}$ is independent of $\Gamma$, then $\sigma_{u,\text{obs}} \propto \Gamma$. Consistent with this argument, (13) reduces to

$$L_{\text{MM}} \approx \frac{\sigma_{u,\text{obs}}}{dU/dz}\text{,} \tag{15}$$

which is also evident inserting (14) into (10). Using (15), $L_{\text{MM}}$ can simply be diagnosed from typical measurements, e.g. 10-minute average cup-anemometer output, at two (or more) heights. The length $L_{\text{MM}}$ can also be cast in terms of variables commonly used in wind engineering, notably the turbulence intensity $I_u$ and shear exponent $\alpha$. Invoking $dU/dz = \alpha U/z$ (Kelly et al., 2014a) and defining $I_{\text{obs}} \equiv \sigma_{u,\text{obs}}/U$, then (15) becomes $L_{\text{MM}} \approx z I_{\text{obs}}/\alpha.$

[revised manuscript text omitted]
_{MM}$ closer to the spectrally-diagnosed $L_{MM}$, and within 10% of $P(L_{MM})$ over a range of $L_{MM}$ from below 10 m to beyond 100 m. It should also be noted that  ignoring speeds below 7 m s$^{-1}$
* * *
[9]The spectral fits were done using spectral-tensor model output over the parameter ranges of $5 < L_{MM} < 500$ m and $0 \leq \Gamma \leq 5$. Some spectra were poorly fitted; since these occurred when $\Gamma=5$, cases with $\Gamma>4.95$ were excluded from the analysis here. As justification, I note that only a small fraction of the cases ($< 10\%$) had such $\Gamma$, and that we only consider well-fit spectra for reliable comparison of parameters.

[Figure]

**Figure 3.** Joint probability density function of predicted and diagnosed (observed) turbulent length scale, from measurements at Høvsore over the homogeneous eastern land sectors. "$x$-axis": Mann-model scale $L_{\mathrm{MM}}$ from spectral fits; "$y$-axis": $L_{\mathrm{MM}}$ estimated from direct measurements of $dU/dz$ and $\sigma_u$, via (15).

can lead to slightly  smaller $L_{\mathrm{MM}}$, since these low wind speeds are more influenced by unstable conditions. Indeed for $L_{\mathrm{MM}} \gtrsim 50\,\mathrm{m}$, including the lower wind speeds  causes both diagnosed and predicted $L_{\mathrm{MM}}$ to increase roughly 10%; this is consistent with larger turbulent eddies being created under unstable conditions.

[Figure]

**Figure 4.** Probability density function of turbulent length scale from observations at Høvsøre from the homogeneous eastern land sectors. Black: Mann-model scale from fits to spectra; dotted/blue: 'mixing-length' formulation ($\ell_m \propto u_*/|dU/dz|$) with revised constant; dashed/gold: Peña et al. (2010) form for $\ell_m$; red/long-dashed: $\sigma_u/|dU/dz|$ form (15).

**3.2.1 Estimating $P(L_{MM})$ in coastal/offshore conditions**

To demonstrate the (probabilistic) use of (13) or (15) for $L_{MM}$ in somewhat different conditions, we now consider flow from offshore, using data from the same mast and height as above (Høvsøre, $z = 80$ m) but for wind directions between $240°$ and $300°$. The mast is roughly 1.75 km east of the coastline and subsequently 1.65 km east of a 16–17 m-high sand dune that lies 100 
[revised manuscript text omitted]